Wing bone laminarity is not an adaptation for torsional resistance in bats

Lee Andrew H. alee712@gmail.com
Simons Erin L.R.
Department of Anatomy, Midwestern University , Glendale, AZ , USA
Thewissen J.
Electronic publication date: 2015 Mar 5
Publication date: 2015
Volume: 3
Electronic Location ID: e823
Received 2014 Jul 25; Accepted 2015 Feb 16
Copyright: © 2015 Lee and Simons
Copyright year: 2015
Copyright holder: Lee and Simons
License: This is an open access article distributed under the terms of the Creative Commons Attribution License, which permits unrestricted use, distribution, reproduction and adaptation in any medium and for any purpose provided that it is properly attributed. For attribution, the original author(s), title, publication source (PeerJ) and either DOI or URL of the article must be cited.
License URL: https://creativecommons.org/licenses/by/4.0/

Keywords: Flight, Bone histology, Avascular bone, Growth rate, Bat, Bird, Field metabolic rate, Cross-sectional geometry

Funding: Midwestern University Funding for this work was provided by start-up grants to Andrew H. Lee and Erin L.R. Simons from Midwestern University. The funders had no role in study design, data collection and analysis, decision to publish, or preparation of the manuscript.

==============================
Torsional loading is a common feature of skeletal biomechanics during vertebrate flight. The importance of resisting torsional loads is best illustrated by the convergence of wing bone structure (e.g., long with thin walls) across extant bats and birds. Whether or not such a convergence occurs at the microstructural level is less clear. In volant birds, the humeri and ulnae often contain abundant laminar bony tissue in which primary circumferential vascular canals course concentrically about the long axis of the bone. These circumferential canals and the matrix surrounding them presumably function to resist the tissue-level shear stress caused by flight-induced torsion. Here, we assess whether or not laminar bone is a general adaptive feature in extant flying vertebrates using a histological analysis of bat bones. We sampled the humeri from six adult taxa representing a broad phylogenetic and body size range (6–1,000 g). Transverse thick sections were prepared from the midshaft of each humerus. Bone tissue was classified based on the predominant orientation of primary vascular canals. Our results show that humeri from bats across a wide phylogenetic and body size range do not contain any laminar bone. Instead, humeri are essentially avascular in bats below about 100 g and are poorly vascularized with occasional longitudinal to slightly radial canals in large bats. In contrast, humeri from birds across a comparable size range (40–1,000 g) are highly vascularized with a wide range in bone laminarity. Phylogenetically-informed scaling analyses reveal that the difference in vascularity between birds and bats is best explained by higher somatic relative growth rates in birds. The presence of wing bone laminarity in birds and its absence in bats suggests that laminar bone is not a necessary biomechanical feature in flying vertebrates and may be apomorphic to birds.

Introduction

Flapping flight only evolved in three vertebrate clades: pterosaurs, birds, and bats. In each clade, the forelimb was modified into an airfoil using a different combination of skeletal elements and integument. Yet in these clades, the repeated evolution of proximal wing bones with large diameters and thin walls highlights similar biomechanical constraints to optimize bending and torsional (i.e., twisting) resistance (Currey & Alexander, 1985; Cubo & Casinos, 1998). In vivo demonstration of bone strains confirms substantial torsional loading in the humerus of a large fruit bat and the pigeon (Swartz, Bennett & Carrier, 1992; Biewener & Dial, 1995). Although generalization from limited interspecific sampling of in vivo bone strains warrants caution, consilience from bone geometry and kinetics suggests that amniotes evolved gross skeletal features to resist torsional loads during flapping flight (Swartz, Bennett & Carrier, 1992; Biewener & Dial, 1995).

The biomechanical influence of flapping flight may also govern organization at the histological level. One of the histological features of bone under investigation is the network of primary vascular canals, which are small channels containing blood vessels, loose connective tissue, and visceral nerves (Ross & Pawlina, 2011). Primary vascular canals vary in orientation across the adult limb skeleton (Fig. 1) and are well-documented particularly in birds (e.g., de Margerie, 2002; de Margerie, Cubo & Castanet, 2002; Skedros & Hunt, 2004; de Margerie et al., 2005; Simons & O’Connor, 2012). In particular, primary vascular canals oriented circumferentially to form laminar bone (Fig. 1A) are abundant in the midshaft humerus. Because this location is hypothesized to experience elevated torsional loading during flight, the presence of laminar bone in the humerus may relate to a biomechanical function (de Margerie, 2002; de Margerie et al., 2005). Wing shape may also influence the proportion of laminar bone in the humerus (laminarity index sensu de Margerie, 2002) presumably because broad wings produce greater twisting moments (torques) than narrow wings (de Margerie et al., 2005; Simons & O’Connor, 2012). Together, these findings suggest that laminar bony tissue may be a biomechanical adaptation to flight-induced torsional loads at least in birds (de Margerie, 2002; de Margerie et al., 2005).

Figure 1 Variation in vascular canal orientation with skeletal element and growth rate in the chicken.

As seen in transverse section, the humerus, ulna, and femur (A) contain abundant laminar bone, which consists mainly of circumferential vascular canals. The radius (B) shows predominately longitudinal vascular canals. A femur from an individual selected for rapid growth (C) contains an abundance of radial canals. In each panel, periosteal surface points up. Scale bars = 200 µm.

The convergence of flapping flight in bats (Norberg, 1994) presents an opportunity to address the extent of laminar bone as a general histological response to flight-induced torsion. Yet, there are few studies detailing bone histology in bats. The most taxonomically comprehensive of these is limited to the femur, which generally consists of lamellar bony tissue and shows size-dependent vascularity; large-bodied taxa have poorly vascularized femora with predominantly longitudinal primary vascular canals, whereas small-bodied taxa have avascular femora (Foote, 1916). At least for three taxa, various proximal wing elements are also poorly vascularized or avascular (Enlow & Brown, 1958; de Ricqlès et al., 1991; Papadimitriou, Swartz & Kunz, 1996; Bennett & Forwood, 2010). However, data from the humerus are limited altogether to a single small-bodied species of undetermined taxonomic identity (Enlow & Brown, 1958). Here, we present a more comprehensive assessment of the histology of bat humeri to test for the presence of laminar bony tissue. As a basis for comparison, we also present histology of bird humeri from available volant taxa. If patterns of bone laminarity in bats and birds converge upon each other, then there may be a fundamental torsion-resisting constraint at the tissue level.

Alternatively, the humeri of bats may be avascular or poorly vascularized. To test whether bone vascularity (or its absence) influences whole-bone torsional rigidity, we compare how polar section modulus (a measure of cortical distribution) scales with body size in bats and birds. Because the proximal wing elements of volant amniotes are generally thin-walled (Currey & Alexander, 1985; Swartz, Bennett & Carrier, 1992), we expect the torsional rigidity of bat and bird humeri to scale similarly in spite of potential differences in bone vascularity. This would suggest that adaptation of the humerus to flight-induced torsion is principally in bone shape rather than in tissue-level organization.

Other factors besides biomechanics may contribute to differences in bone vascularity between bat and bird humeri. As conduits for blood vessels (Ross & Pawlina, 2011), vascular canals may promote molecular exchange (Seymour et al., 2012). Therefore, we expect bone vascular patterns to correlate directly with physiological factors that depend on molecular exchange. Two such factors that have received attention are somatic rates of active (field) metabolism (e.g., Seymour et al., 2012) and growth (e.g., Amprino, 1947; de Margerie, Cubo & Castanet, 2002; Williams et al., 2004). To test which of these two factors may account for differences in bone vascularity, we look for corresponding differences in chiropteran and avian scaling of field metabolic rate and somatic maximum growth rate. By disentangling mechanical, metabolic, and growth-related influences on the histology of bat and bird humeri, we gain insight into the factors that govern skeletal variation in flying vertebrates more generally.

Materials & Methods

Sampling and histology

We collected humeri from six extant chiropteran taxa (Table 1). The sampled taxa represent both major clades of bats (Teeling et al., 2005; Hutcheon & Kirsch, 2006; Tsagkogeorga et al., 2013), Yangochiroptera (Vespertilioniformes) and Yinpterochiroptera (Pteropodiformes) and span three orders of magnitude in body mass (6–1,000 g), representing nearly the entire size range found in bats (Norberg & Norberg, 2012). Whereas the sampled bats vary in specific feeding behaviors (e.g., insectivore, frugivore, piscivore, nectarivore), they have relatively similar wing shapes (aspect ratios ≈6–9: Norberg & Rayner, 1987). We also sampled several avian taxa with broad wing shape similar to the bats as part of our ongoing studies on wing bone laminarity and include some of those data here for comparison (Table 1). To minimize the confounding effects of ontogeny on bone histology (e.g., Padian, 2013), only skeletally mature specimens were sampled. Prior to histological sampling, length of each humerus was measured (Table 1), and three-dimensional replicas were produced by molding and casting (Lamm, 2013).

Table 1 Properties of analyzed specimens.

Taxon	Specimen	Mass
(g)	Humeral
length
(mm)	LI
(95%CI)	Zp
(mm3)	
Bats						
Rhinolophus lepidus	pers. coll. H1-2	6	24	0	0.163	
Macrotus californicus	UA 3767 H1-2	12	28	0	0.284	
Myotis myotis	(Meier et al., 2013)	29	33	n/a	0.435	
Phyllostomus discolor	UA 16197 H1-1	34	35	0	0.921	
Noctilio leporinus	UA 15743 H1-2	61	42	0	2.426	
Rousettus leschenaultii	pers. coll. H1-2	108	54	0	3.312	
Pteropus vampyrus	pers. coll. H1-2	1024	129	0	30.586	
Birds						
Oceanodroma tethys	NMNH 614194	23	28	n/a	0.627	
Oceanites oceanicus	CMNH 7752	29	22	n/a	0.790	
Phalaenoptilus nuttallii	MWU 264 H1-1	48	32	0.259 (0.082)	1.156	
Bulweria bulwerii	NMNH 556263	99	59	n/a	2.265	
Nothura darwinii	UF 22260 H1-2	274	43	0.316 (0.055)	4.762	
Crypturellus boucardi	UF 44840 H1-1	418	48	0.211 (0.057)	10.238	
Crypturellus cinnamomeus	UA 8699 H2-1	422	50	0.388 (0.047)	12.156	
Columba livia	MWU 256 H1-2	455a	49	0.063 (0.024)	11.801	
Nothoprocta cinerascens	UF 38951 H1-2	480a	59	0.462 (0.048)	12.921	
Calonectris diomedea	OUVC 10438	535	125	0.070	28.887	
Nothocercus nigrocapillus	UF 43432 H1-2	605a	60	0.115 (0.037)	11.896	
Eudromia elegans	UF 22257 H1-1	680	62	0.578 (0.052)	14.756	
Tinamus major	UF 44828 H1-2	960	73	0.316 (0.044)	34.868	
Buteo jamaicensis	OUVC 10506	1126	109	0.703	50.879	
Anhinga anhinga	OUVC 10432	1235	126	0.233	39.635	
Phalacrocorax auritus	OUVC 10482	1960	144	0.390	39.449	
Cathartes aura	OUVC 9648	2006	160	0.376	121.185	
Pelecanus occidentalis	OUVC 10484	3438	273	0.478	163.645	
Notes.

CMNH Carnegie Museum of Natural History

MWU Midwestern University

NMNH Smithsonian Institution National Museum of Natural History

OUVC Ohio University Vertebrate Collections

UA University of Arizona

UF Florida Museum of Natural History

LI laminarity index

a known mass of individual; Zp, polar section modulus.

A mid-diaphyseal block was removed from each humerus using a rotary tool (Dremel 4000; Dremel, Mt. Prospect, Illinois, USA). The following steps are summarized in Table S1. Blocks of bone were fixed under vacuum in 10% neutral buffered formalin. After fixation, the blocks were successively dehydrated using 70%, 85%, and 100% ethanol under vacuum. The blocks were cleared under vacuum with a xylene-substitute (Histo-Clear; National Diagnostics, Atlanta, Georgia, USA). Infiltration of the blocks was performed by vacuum-impregnation in a series of methyl methacrylate (MMA) resins (Osteo-Bed, Polysciences Inc., Warrington, Pennsylvannia, USA). To complete embedding, each bone block was transferred to an air-tight glass vial filled with MMA. Vials were placed in a bead bath set to 32 °C and allowed 48 h to fully harden.

The plastic-embedded specimens were released from the glass vials, and two transverse 700-µm wafers were cut from each specimen at midshaft using a precision saw (Isomet 1000; Buehler, Lake Bluff, Illinois, USA). Wafers were glued (2-Ton epoxy; Devcon, Milpitas, California, USA) to frosted glass slides and manually ground (Metaserv 250; Buehler, Lake Bluff, Illinois, USA) to 100 ± 10 µm.

Imaging

Whole thick sections were imaged without coverslips under non-polarized light to measure vascular canal orientation as well as under circularly polarized light to distinguish between parallel-fibered and woven bone matrix (Figs. S1 and S2). After polarized light microscopy was performed, sections were acid-etched and stained with a heated solution of toluidine blue (Eurell & Sterchi, 1994). This particular preparation of toluidine blue does not require the removal of embedding resin and works on thick sections already mounted to slides. Because this base-like metachromatic dye highlights features rich in acidic or anionic components such as cement lines of secondary osteons, medullary bone, and hyaline cartilage, we were able to quickly identify and exclude vascular canals of secondary osteons from further analysis (Figs. S3 and S4). Stained and unstained sections were imaged using a motorized light microscope (Ni-U; Nikon, Tokyo, Japan, USA) with a strain-free long working distance objective (10× Plan Fluor: numerical aperture of 0.3, resolvable size ≈1 µm). Focus and stitching of histological montages were controlled by software (NIS Elements D; Nikon, Tokyo, Japan, USA). The montages were sharpened using Photoshop (CS5; Adobe, San Jose, California, USA) with the “Unsharp Mask” filter set at 10 px and are high resolution (2.1 µm per pixel). In accordance with major grant funding agencies and to promote data transparency, the montages are freely accessible as interactive digital slides at the Paleohistology Repository (Lee & O’Connor, 2013: http://paleohistology.appspot.com). After imaging, a standard glass coverslip was mounted (Permount; Fisher Scientific, Hampton, New Hampshire, USA) to each physical slide to arrest long-term cracking of undecalcified bone sections.

Assessing bone laminarity

A method to measure primary vascular canal orientation was first proposed by de Margerie (2002) and later used by subsequent studies (e.g., Lee, 2004; Skedros & Hunt, 2004; Cubo et al., 2005; Simons & O’Connor, 2012). Although we recognize that this method was modified to offer greater descriptive precision (de Boef & Larsson, 2007; Lee, 2007), the added precision is excessive for the bat dataset because canal orientation is unambiguous. Therefore, we used the method by de Margerie (2002) with slight modifications to improve throughput.

Using Photoshop, we divided the cortex of each thick section into octants and extracted the four cortical octants representing cardinal anatomical positions (bats: anterior, posterior, lateral, and medial; birds: cranial, caudal, dorsal, and ventral). We avoided measurement below the plane of section (i.e., the polished surface) and only measured the most in-focused portions of each primary vascular canal. If a canal branched, it was separated into individual canals at the node. Vascular canals of secondary osteons were excluded from further analysis. Using ImageJ, an ellipse was fitted inside each canal, and aspect ratio and orientation of the major axis relative to the periosteal surface were measured. To determine canal orientation from the aspect ratio and major axis of the ellipse, we followed criteria proposed by de Margerie (2002). The criteria are summarized as follows: (1) longitudinal canals have an aspect ratio of less than 3 (Fig. 1B); (2) circumferential canals have major axes oriented 0° ± 22.5° relative to the nearest tangent line drawn at the periosteal surface (Fig. 1A); (3) radial canals have major axes oriented 90° ± 22.5° relative to the nearest tangent line drawn at the periosteal surface (Fig. 1C); and (4) all remaining canals are oblique.

In most sections, the curvature of the periosteal surface means that the tangent line needed to measure circumferential and radial canals changes across the cortical octant. The repeated referencing of each canal to a variable periosteal tangent line is time-consuming and increases the likelihood of measurement error. We standardized the tangent line and thereby increased the throughput of canal measurement by uncurving each cortical octant relative to the periosteal surface using the “Straighten” function in ImageJ (1.49b, National Institutes of Health, USA). Once a cortical quadrant is “straightened”, the periosteal surface is aligned with the horizontal (Figs. 2 and 3) and a constant reference line is established. To assess the extent of image deformation on canal orientation, we overlaid a graphical layer containing known test angles relative to the periosteal surface and circular profiles prior to straightening. On average, image deformation is acceptable; test angles deviate no more than 5 degrees from their original values, and circular profiles maintain aspect ratios between 1.00–1.17. At least for the currently sampled bones with relatively thin cortices, the “Straighten” function preserves original orientation data.

Figure 2 Histology of humeri in sampled bats ordered by ascending body mass.

Non-polarized imagery reveals avascular to poorly vascularized compacta. Representative views are from the anterior octant of (A) Rhinolophus lepidus, (B) Macrotus californicus, (C) Phyllostomus discolor, (D) Noctilio leporinus, (E) Rousettus leschenaultii, and (F) Pteropus vampyrus. Periosteal surface points up in each panel. Scale bar equals (A) 150 µm, (B) 200 µm, (C) 300 µm, (D & E) 400 µm, and (F) 800 µm. Digital slides are available at http://paleohistology.appspot.com.

Figure 3 Histology of humeri in sampled birds ordered by ascending body mass.

Representative views are from the cranial octant of (A) Phalaenoptilus nuttallii, (B) Nothura darwinii, (C) Crypturellus boucardi, (D) Crypturellus cinnamomeus, (E) Columba livia, (F) Nothoprocta cinerascens, (G) Nothocercus nigrocapillus, (H) Eudromia elegans, and (I) Tinamus major. Periosteal surface points up in each panel, and a noticeable external fundamental system occurs in both (G) and (I). Scale bar equals (A) 300 µm, (B, C, & E) 400 µm, (F & G) 480 µm, (D & I) 600 µm, and (H) 800 µm. Digital slides are available at http://paleohistology.appspot.com.

We calculated the proportion of circumferential canals for sections with a non-zero number of circumferential canals (laminarity index sensu de Margerie, Cubo & Castanet, 2002) using the Wilson estimate for a population proportion and confidence interval. This estimate accounts for bias from small sample sizes (Samuels & Witmer, 1999). Published laminarity index values from six individuals of broad-winged birds were included for comparison (Simons & O’Connor, 2012).

Compilation of relative growth and field metabolic rates

We compiled chiropteran and avian growth curves (40 and 56, respectively) from available literature (Zullinger et al., 1984; Kunz & Stern, 1995; Starck & Ricklefs, 1998; Koehler & Barclay, 2000; Kunz & Hood, 2000; McLean & Speakman, 2000; Jin et al., 2012). For each species, two values were calculated at the age of growth inflection (see equations in: Lee et al., 2013): (1) maximum growth rate and (2) body mass (Table S3). Maximum growth rate is strongly correlated with mass (Case, 1978). To account for the confounding effect of size on maximum growth rate, relative growth rate (RGR) was used instead and was calculated as the ratio of maximum growth rate and mass at inflection (e.g., Cooper et al., 2008).

Field metabolic rates and mass representing species-averages of 14 bats and 86 birds (Table S4) were compiled from previous studies (Delorme & Thomas, 1999; Geiser & Coburn, 1999; Nagy, Girard & Brown, 1999; Voigt, Kelm & Visser, 2006; Hudson, Isaac & Reuman, 2013). Most of these studies used the “doubly labeled water” method (see review: Nagy, 2005) in free-ranging individuals to estimate field metabolic rate (FMR). In contrast, the study by Delorme & Thomas (1999) estimated FMR from individuals of Artibeus jamaicensis and Rousettus aegyptiacus that were allowed to feed freely but not to fly. While confined, these individuals did not experience major locomotor costs associated with flight or weight gain. Yet, FMR estimates from the confined bats are close to mean values expected in free-ranging bats of comparable size. Therefore, the inclusion of the FMR estimates from these confined bats is reasonable. Moreover, they are needed to increase the modest sample size and phylogenetic breadth of the chiropteran FMR dataset. To remove the confounding effects of size on FMR, estimates were standardized by mass. The resulting values of mass-specific FMR and RGR were log10-transformed before use in regression analyses.

Cross-sectional geometry

In Photoshop, we prepared cortical bone profiles of the humeral sections by tracing the periosteal and endosteal surfaces of each histological montage and filling the area of the cortical bone white and the background black. To supplement our sample, we included a published cortical bone profile of the humerus from the bat Myotis myotis (Meier et al., 2013) as well as profiles scanned by microCT representing nine avian taxa (Simons, Hieronymus & O’Connor, 2011; ELR Simons, 2008, unpublished data). Cortical bone profiles were processed using BoneJ for ImageJ to calculate the cross-sectional geometric parameters of each specimen (Table 1). For this study, we primarily focused on the polar section modulus (Zp), which is an estimate of torsional rigidity and average bone bending rigidity (Ruff, 2002). The torsional rigidity of a long bone is related to an applied torque (force and cross-sectional radius), material properties, length, and the geometric property called the torsion constant. When the cross-sectional profile of a long bone is nearly circular, the torsion constant is equal to the polar moment of area (J) (Craig, 2000). The same is true for Zp, which is J standardized by the cross-sectional radius (i.e., the moment arm). Because Zp is a better estimate of torsional rigidity when the transverse section of a bone is circular, we used a shape ratio (Imax/Imin: Table S2) to determine the degree of circularity (Daegling, 2002). Caution is recommended when a section is strongly elliptical, but even in a non-circular section, Zp can still be a heuristic estimate of overall torsional and bending rigidity (Ruff et al., 2013). To compare the torsional rigidity of humeri across bats and birds of different body and wing sizes, we regressed Zp against the product of body mass and total length of the humerus (e.g., Ruff, 2000; Stock & Shaw, 2007; Young, Fernández & Fleagle, 2010). Specimens without associated body mass data were assigned published mean values for the species (Kunz & Stern, 1995; Savage et al., 2004; Dunning, 2008; de Magalhaes et al., 2009). Values were log10-transformed before use in regression analysis.

Phylogenetically informed scaling analyses (PISA)

Ordinary least squares (OLS) regression, which is commonly used in comparative scaling analyses, assumes that data points are statistically independent. This assumption, however, often is violated in comparative datasets because data points of closely related species are more likely to be similar to each other than to those of distantly related species. By failing to account for this phylogenetic pseudoreplication, an OLS analysis can either detect a significant trend where none exists (i.e., confidence intervals are overly narrow) or incorrectly specify the directionality of a significant trend (Nunn & Barton, 2001).

Recent advances in mammalian and avian systematics make it possible to incorporate phylogenetic information into scaling analyses of polar section modulus (Zp), relative growth rate (RGR), and mass-specific field metabolic rate (FMR). The most recent phylogenies of bats (Agnarsson et al., 2011; Tsagkogeorga et al., 2013) and supertree of birds (Davis & Page, 2014) either present a time-calibrated phylogeny of a small subset of sampled taxa or are completely uncalibrated to geological time. Therefore, we used older time-calibrated supertrees of mammals (Bininda-Emonds et al., 2007) and birds (Jetz et al., 2012). Both supertrees were pruned to match our taxonomic sampling of bats and birds, respectively, in R (R Development Team, 2014) with the “ape” package (Paradis, Claude & Strimmer, 2004). Polytomies were broken and replaced with zero-length branches (Martins & Hansen, 1996).

We performed phylogenetically informed scaling analyses (PISA) in a phylogenetic generalized least-squares (PGLS) framework (Martins & Hansen, 1997; Garland & Ives, 2000) as implemented by the “ape” package. Variance–covariance matrices were calculated to approximate Brownian motion as well as an Orstein-Uhlenbeck model of evolution (Martins & Hansen, 1997) using “corsBrownian” and “corsMartin,” respectively. Variance–covariance structures were passed into the R package “nlme,” and regressions were fit by maximizing the restricted log-likelihood. Model I regression was used because model II regression tends to overcorrect when a dataset does not have enough within-species replication to estimate error variances (Carroll & Ruppert, 1996; Carroll et al., 2006). The resulting OLS and PGLS model I regressions were compared on the basis of three criteria. First, we tested each trait for the presence of significant phylogenetic signal (Blomberg & Garland, 2002) using the R package “picante” (Kembel et al., 2010). If we could not detect significant phylogenetic signal (this does not imply the absence of phylogenetic signal in the trait), we selected the OLS model. Second, we assessed the residual plots of each model for randomness to assess the appropriateness of a linear model for log-transformed data. Third, we evaluated the remaining models and chose the one with the lowest value of the corrected Akaike’s Information Criterion (AICc: Hurvich & Tsai, 1989).

Results and Discussion

Bat humeri do not consist of laminar bone

Across both major clades of bats, Vespertilioniformes and Pteropodiformes, small- to medium-bodied taxa (∼6–100 g) have essentially avascular humeri (Figs. 2A–2E). In contrast to the avascularity found in smaller forms, the humerus of the large-bodied pteropodiform Pteropus vampyrus (∼1,000 g: Kunz & Jones, 2000) is sparsely vascularized (Fig. 2F). Where present, primary vascular canals have a prominent longitudinal to slightly radial orientation, which is inconsistent with the hypothesis that wing bones habitually loaded in torsion contain predominantly circumferential canals. Indeed, laminar bone is completely absent in the humeri of sampled bats (Table 1). Because the sample represents a wide range in phylogeny, feeding behavior, and body size, the likelihood is strong that bat humeri do not contain laminar bone.

The vascular dichotomy between small- and large-bodied bats extends beyond the humerus to other limb elements. In the radius, a small bat (of undetermined taxonomic identity: Enlow & Brown, 1958) shows avascular tissue, whereas a large bat (Pteropus poliocephalus at 750 g: Bennett & Forwood, 2010) has weakly vascularized bone with longitudinal canals. Moreover, in the femur, vascularized bone with longitudinal to slightly radial canals appears restricted to species of large body size such as those of Pteropus (i.e., molossinus, aldabrensis, lepidus (cf. hypomelanus: Jones & Kunz, 2000), and poliocephalus) (Foote, 1916). The histology of the femur is remarkably similar to that of the humerus in bats of comparable class size, a pattern seen in some avian species as well (de Margerie, 2002; Marelli & Simons, 2014).

Vascularity of bat humeri relates to a threshold in adult body size

Although poorly vascularized, the humerus of Pteropus vampyrus is still more vascularized than those of the other bats in the sample. This difference in vascularity seems unrelated to wing shape. Noctilio leporinus and P. vampyrus have similar wing aspect ratios (9.0 and 8.4, respectively: Norberg & Rayner, 1987) yet differ in vascularity of the humerus (Figs. 2D and 2F). Feeding behavior also is an unlikely explanation. Frugivory characterizes the primary feeding behavior of both Rousettus leschenaultii and P. vampyrus (Norberg & Rayner, 1987), but they have humeri with contrasting vascularity (Figs. 2E and 2F).

Across amniotes, maximum somatic growth rates (Case, 1978) and field metabolic rates (Nagy, 2005) are strongly dependent on body mass. Consequently, the increased bone vascularity in large-bodied bats compared to small-bodied bats may reflect rapid somatic growth rate, elevated field metabolic rate, or large size. When the confounding influence of size on somatic growth and field metabolic rate is accounted for (i.e., a relative growth rate and mass-specific field metabolic rate, respectively), large-bodied bats have relatively low growth and field metabolic rates compared to small-bodied ones (Figs. 4 and 5). In both growth and metabolic datasets, significant negative scaling trends are revealed by OLS and PGLS regressions (Tables 2 and 3). Moreover, scaling relationships recovered by OLS and PGLS regressions in the growth and metabolic datasets, respectively, are statistically indistinguishable. Consilience from several models of evolution suggests that neither growth rate nor field metabolic rate alone explains the increased bone vascularity (humerus or otherwise) in large-bodied bats. Instead, there appears to be a threshold of adult size at ∼100–200 g above which bats show vascularized humeri (Fig. 2F), radii (Bennett & Forwood, 2010) and femora (Foote, 1916). Other clades of amniotes also show a size threshold in bone vascularity, although the precise threshold varies with the clade (de Margerie et al., 2005; de Buffrénil, Houssaye & Böhme, 2008; Werning, 2013). Large size in bats and other amniotes may necessitate vascularized cortical bone to provide additional capacity for nutrient and waste exchange beyond that supplied by canaliculi (de Ricqlès et al., 1991).

Figure 4 Interspecific scaling of mass-specific field metabolic rate (FMR) in bats and birds.

Bat and bird data points are omitted for clarity. Shaded regions are 95% confidence bands. Representative views of histology from avian ((A) Phalaenoptilus nuttallii, (B) Nothura darwinii, (C) Tinamus major) and chiropteran ((D) Phyllostomus discolor, (E) Rousettus leschenaultii, (F) Pteropus vampyrus) humeri suggest that the vascular dichotomy between bats and birds is not related to mass-specific FMR.

Table 2 Scaling of log10(mass) and log10(mass-specific field metabolic rate).

Taxon	Model	Slope	95% CI	Intercept	95% CI	
Bats (n = 14)	OLS (AICc = −7.0; random residuals)	−0.358	−0.515, −0.201	0.938	0.702, 1.173	
	PGLS-BM (AICc = 7.1; random residuals)	−0.405	−0.615, −0.195	0.996	0.564, 1.427	
	PGLS-OU (α = 1.1; AICc = 3.8; random residuals)	−0.358	−0.499, −0.217	0.938	0.725, 1.150	
	PGLS-OU (α = 1.8; AICc = 3.8; random residuals)	−0.358	−0.499, −0.217	0.938	0.725, 1.150	
Birds (n = 86)	OLS (AICc = − 44.5; random residuals)	−0.356	−0.406, −0.305	1.066	0.967, 1.164	
	PGLS-BM (AICc= −73.7; random residuals)	−0.364	−0.444, −0.284	1.002	0.732, 1.271	
	PGLS-OU (α = 0; AICc = − 72.0; random residuals)	−0.367	−0.446, −0.289	1.019	0.546, 1.491	
	PGLS-OU (α = 10; AICc = − 30.9; random residuals)	−0.356	−0.405, −0.306	1.066	0.969, 1.163	
Notes.

Abbreviations follow text. Models selected for interspecific comparison in bold.

Table 3 Scaling of log10(asymptotic mass) and log10(somatic relative growth rate).

Taxon	Model	Slope	95% CI	Intercept	95% CI	
Bats (n = 40)	OLS (AICc = 8.3; random residuals)	−0.452	−0.590, −0.314	−0.852	−1.070, −0.634	
	PGLS-BM (AICc = 10.1; non-random residuals)	−0.258	−0.455, −0.062	−1.169	−1.564, −0.775	
	PGLS-OU (α = 0.02; AICc = 9.3; non-random residuals)	−0.306	−0.491, −0.120	−1.099	−1.419, −0.779	
	PGLS-OU(α =1.5; AICc= 18.6; random residuals)	−0.452	−0.585, −0.319	−0.852	−1.063, −0.641	
Birds (n = 56)	OLS (AICc = − 46.2; random residuals)	−0.330	−0.384, −0.276	−0.227	−0.329, −0.126	
	PGLS-BM (AICc = − 58.5; non-random residuals)	−0.290	−0.364, −-0.215	−0.425	−0.675, −0.174	
	PGLS-OU (α = 0; AICc = − 57.4; non-random residuals)	−0.287	−0.358, −0.217	−0.418	−0.756, −0.081	
	PGLS-OU (α= 2.5;AICc = −32.8; random residuals)	−0.330	−0.383, −0.277	−0.227	−0.327, −0.128	
Notes.

Abbreviations follow text. Models selected for interspecific comparison in bold.

Bird humeri have variable bone laminarity despite habitual flight-induced torsion

In contrast to bats, volant birds sampled across a similar body size range (40–1,000 g) have noticeably vascularized humeri (Fig. 3). Overall, primary vascular canal orientation as represented by the laminarity index is moderately circumferential (Table 1). However, low laminarity (0.063) in the humerus of Columba livia (Fig. 3E) is counterintuitive given that the humerus in this species is known to experience high torsional loads during flight; torsional shear strains are 1.5 times greater than longitudinal strains (Biewener & Dial, 1995). This result weakens the correlation between laminar bone and torsional loading and suggests the need for increased sampling and reassessment by future studies. Put together, low laminarity in a pigeon humerus, ubiquity of essentially avascular humeri in small- to medium-bodied bats, and longitudinal orientation of canals in the humerus of one of the largest species of bat suggest that laminar bone is not a necessary biomechanical adaptation in flying vertebrates.

Adult body size does not explain vascular dichotomy between bats and birds

Bats and birds do not share the same adult size threshold for bone vascularity. Avian taxa as small as 48 g (Phalaenoptilus nuttallii: Fig. 3A) contain abundant vascular canals in the humerus in striking contrast to bats of comparable size (e.g., Phyllostomus discolor and Noctilio leporinus: Figs. 2C and 2D). Furthermore, large body size does not reduce the vascular disparity between the humeri of bats and birds. At approximately 1,000 g, Pteropus vampyrus (Fig. 2F) has a humerus with far fewer canals (let alone circumferentially oriented ones) than Tinamus major (Fig. 3I). Clearly, adult body size is not responsible for the strong dichotomy in vascularity between bat and bird humeri.

Torsional rigidity does not explain vascular dichotomy between bats and birds

Notwithstanding the vascular dichotomy, humeri in adult bats and birds of comparable size may have similar cross-sectional geometry. Specifically, we are unable to reject the possibility that chiropteran and avian polar section modulus (Zp) scales isometrically with the product of mass and humeral length (Fig. 6). Scaling relationships based on OLS and PGLS regression are indistinguishable (Table 4) probably because the sample size of seven bats and 18 birds is too small for the effects of phylogeny to be significant. Of course, greatly increased sampling may improve detection of differences in how Zp scales with size between chiropteran and avian humeri. There is, however, independent evidence to support the interpretation that some mechanical properties of bat and bird humeri really are similar.

Table 4 Scaling of log10(polar section modulus of humerus) and log10(body mass x humeral length).

Taxon	Model	Slope	95% CI	Intercept	95% CI	
Bats (n = 7)	OLS (AICc=−1.8; random residuals)	0.788	0.688, 0.889	−2.599	−2.941, −2.257	
	PGLS-BM (AICc = 23.9; random residuals)	0.769	0.687, 0.851	−2.550	−2.834, −2.266	
	PGLS-OU (α = 0; AICc = 25.9; random residuals)	0.771	0.689, 0.854	−2.556	−3.016, −2.097	
	PGLS-OU (α = 1; AICc = 26.1; random residuals)	0.789	0.712, 0.865	−2.599	−2.860, −2.338	
Birds (n = 18)	OLS (AICc = −13.5; random residuals)	0.778	0.695, 0.860	−2.397	−2.770,−2.024	
	PGLS-BM (AICc = − 1.6; random residuals)	0.792	0.712, 0.871	−2.445	−2.840, −2.050	
	PGLS-OU (α = 0.3; AICc = − 1.5; random residuals)	0.777	0.701, 0.853	−2.397	−2.742, −2.052	
	PGLS-OU (α = 1; AICc = − 1.5; random residuals)	0.777	0.701, 0.853	−2.397	−2.742, −2.052	
Notes.

Abbreviations follow text. Models selected for interspecific comparison in bold.

At least for the humerus, tissue density (Dumont, 2010) and mineral content (Currey, 1984; Papadimitriou, Swartz & Kunz, 1996; Swartz & Middleton, 2008) are relatively unspecialized across birds and mammals, bats included. This similarity of material properties in volant and non-volant taxa suggests that specializations in shape may be the primary means by which bat and bird humeri resist flight-induced torsion (Dumont, 2010). Indeed, relative cortical thickness of humeri is thinner in bats than in non-volant mammals and converges independently on the relative cortical thickness observed in birds (Currey & Alexander, 1985; Swartz, Bennett & Carrier, 1992). Those previous findings together with our results on cross-sectional geometry suggest that the torsional rigidity of bat and bird humeri may be convergent adaptations to loads imposed by flapping flight (Currey & Alexander, 1985; Swartz, Bennett & Carrier, 1992; Biewener & Dial, 1995). Although torsional rigidity of the humerus appears relatively constrained in bats and birds of comparable size, bone vascularity is strongly divergent. This suggests that the biomechanical factors constraining mineral content and cortical geometry in bats and birds have negligible impact on bone vascularity.

Field metabolic rate does not explain vascular dichotomy between bats and birds

The vascular dichotomy in the chiropteran and avian humerus is not related to differences in mass-specific daily energy expenditure. Note that the daily estimates include but are not limited to the costs of maintenance (basal metabolism) as well as of flight, which is metabolically expensive but may not be equivalent in bats and birds given differences in flight efficiency, kinematics, and ecology found in small-bodied taxa (Winter & von Helversen, 1998; Voigt & Lewanzik, 2011; Muijres et al., 2012). Without a more thorough phylogenetic sampling of the costs of flight in bats, we take the circumspect but still ecologically relevant approach with mass-specific field metabolic rate (FMR). OLS and PGLS regressions recover significant negative trends in mass-specific FMR that are statistically indistinguishable between bats and birds (Table 2 and Fig. 4). These results are consistent with preliminary work based on the OLS regression of maximal metabolic rate (Schmidt-Nielsen, 1984, p. 158) and suggest the possibility that active bats and birds at a given size have similar daily energy expenditure per unit mass. Yet, the humeri of bats and birds have strikingly different vascularity contrary to predictions (Fig. 4). This inconsistency suggests that mass-specific FMR is not a strong contributing factor to the vascular dichotomy in bat and bird humeri.

Somatic growth rate may explain vascular dichotomy between bats and birds

Unlike torsional rigidity and mass-specific FMR, somatic measures of relative growth rate (RGR) may account for the vascular dichotomy between bat and bird humeri. Consilience from OLS and PGLS regressions (Table 3) strongly suggests that body mass in bats, on average, grows about four times slower than in birds of comparable size. This difference in relative growth rate is consistent with predictions from Amprino’s Rule (Amprino, 1947); relatively rapid-growing birds express well vascularized humeri, whereas relatively slow-growing bats express avascular to poorly vascularized humeri (Fig. 5). A potential criticism of these comparisons is that somatic RGR is a crude approximation of local expressions of growth. For example, growth rates of periosteal bone differ across the skeleton (e.g., de Ricqlès et al., 1991; Castanet et al., 2000; de Margerie et al., 2004). Indeed, the wings of volant taxa may grow disproportionately faster than the rest of the body as suggested by positive allometry in wing bone length from a small ontogenetic sample of bats and volant birds (e.g., Carrier & Leon, 1990; Cretekos et al., 2008). Cursory examination of available data on the growth in 32 species of bats, however, reveals that the logistic growth constant K (relative growth rate) of forearm (radius) length is slightly greater than but comparable to that of body mass (Kunz & Hood, 2000). The relative growth rate of the chiropteran humerus in these species is not reported, but it is presumably similar to the radius. In avian studies, growth of proximal wing elements is not typically reported, but if growth is positively allometric like that in the California gull (Carrier & Leon, 1990), then somatic RGR can still be heuristic as a minimum estimate of growth in the humerus; that is, the actual growth disparity between bat and bird humeri may be at least what is implied here (Fig. 5). Therefore, a tentative but reasonable conclusion is that the vascular dichotomy between bat and bird humeri of comparably-sized individuals primarily reflects differences in relative growth rate.

Figure 5 Interspecific scaling of somatic relative growth rate (RGR) in bats and birds.

Shaded regions are 95% confidence bands. Representative views of histology from avian ((A) Phalaenoptilus nuttallii, (B) Nothura darwinii, (C) Tinamus major) and chiropteran ((D) Phyllostomus discolor, (E) Rousettus leschenaultii, (F) Pteropus vampyrus) humeri suggest that the vascular dichotomy between bats and birds may be a reflection of differences in growth rate.

Figure 6 Isometric scaling of polar section modulus (Zp) in humeri of bats and birds.

Shaded regions are 95% confidence bands. Representative views of histology from chiropteran ((A) Phalaenoptilus nuttallii, (B) Nothura darwinii, (C) Tinamus major) and avian ((D) Phyllostomus discolor, (E) Rousettus leschenaultii, (F) Pteropus vampyrus) humeri suggest that the vascular dichotomy has little impact on the torsional rigidity of humeri.

Conclusions

Avascular to poorly vascularized humeri of bats suggest that adaptation to flight need not involve laminar bone. Indeed, despite poor vascularity, bat humeri have potentially equivalent cross-sectional rigidity to torsion as bird humeri. In bats, bone shape rather than tissue-level organization may be the principal mechanism of specialization to loads. The humerus is only vascular, albeit sparsely, in adult bats greater than 100–200 g. This contrast in bone vascularity between small and large bats cannot be explained by feeding behavior, wing aspect ratio, mass-specific field metabolic rate, or somatic relative growth rate suggesting that vascular canals may be necessary beyond a size-threshold to supplement canaliculi in nutrient and waste exchange. Surprisingly, potentially similar daily metabolic expenditure in active bats and birds of comparable size need not involve highly vascularized humeri; poor bone vascularity in bats appears sufficient. The vascular disparity between bat and bird humeri may be related to differences in somatic relative growth rate in which bats grow approximately four times slower than comparably-sized birds. Therefore, bats may grow too slowly to form laminar bone.

At this time, however, we cannot assess whether laminar bone is an expression of early juvenile growth or a direct response to flight loads in the taxa where it evolved (namely in birds). Future investigations (some of which are already in progress) should focus on when laminar bone forms during ontogeny and how it responds to evolutionary acquisitions or losses of flight.

Supplemental Information

Table S1 Bone embedding protocol

Click here for additional data file.

Table S2 Additional geometric properties of humeral sections

Abbreviations: CMNH, Carnegie Museum of Natural History; MWU, Midwestern University; NMNH, Smithsonian National Museum of Natural History; OUVC, Ohio University Vertebrate Collections; UA, University of Arizona; UF, Florida Museum of Natural History; CA, cortical area; TA, total area; Imax, maximum second moment of inertia; Imin, minimum second moment of inertia.

Click here for additional data file.

Table S3 Compilation of maximum somatic growth rates

A taxonomic name with an asterisk is spelled as it appears in mammalian (Bininda-Emonds et al., 2007) and avian (Jetz et al., 2012) tree files. Data were compiled from the following sources: (1) Kunz & Hood (2000); (2) Jin et al. (2012); (3) Zullinger et al. (1984); (4) McLean & Speakman (2000); (5) Kunz & Stern (1995); (6) Koehler & Barclay (2000); and (7) Starck & Ricklefs (1998).

Click here for additional data file.

Table S4 Compilation of field metabolic rates

A taxonomic name with an asterisk is spelled as it appears in mammalian (Bininda-Emonds et al., 2007) and avian (Jetz et al., 2012) tree files. Data were compiled from the following sources: (1) Nagy, Girard & Brown (1999); (2) Voigt, Kelm & Visser (2006); (3) Geiser & Coburn (1999); (4) Hudson, Isaac & Reuman (2013); and (5) Delorme & Thomas (1999).

Click here for additional data file.

Figure S1 Collagen fiber orientation of humeri in sampled bats

Circularly polarized light in cleared sections of standardized thickness reveals lamellar and parallel-fibered bone. Black to white intensity of polarized light corresponds to longitudinally oriented and transversely oriented collagen fibers, respectively. Representative views are from the lateral octant of (A) Rhinolophus lepidus, (B) Macrotus californicus, (C) Phyllostomus discolor, (D) Noctilio leporinus, (E) Rousettus leschenaultii, and (F) Pteropus vampyrus. Periosteal surface points up in each panel. Scale bar equals (A & B) 200 µm, (C) 300 µm, (D) 400 µm, (E) 480 µm, and (F) 1,200 µm. Digital slides are available at http://paleohistology.appspot.com.

Click here for additional data file.

Figure S2 Collagen fiber orientation of humeri in sampled birds

Representative views are from the dorsal octant of (A) Phalaenoptilus nuttallii, (B) Nothura darwinii, (C) Crypturellus boucardi, (D) Crypturellus cinnamomeus, (E) Columba livia, (F) Nothoprocta cinerascens, (G) Nothocercus nigrocapillus, (H) Eudromia elegans, and (I) Tinamus major. Periosteal surface points up in each panel. Scale bar equals (A) 300 µm, (B & E) 480 µm, (C, D, F & G) 600 µm, and (H & I) 800 µm. Digital slides are available at http://paleohistology.appspot.com.

Click here for additional data file.

Figure S3 Bone histology of humeri in sampled bats stained to highlight cement lines

Sections stained with toluidine blue reveal lamellar bone, a complex canalicular network, resorption fronts, and canals of secondary osteons, which were excluded from analysis. Representative views are from the posterior octant of (A) Rhinolophus lepidus, (B) Macrotus californicus, (C) Phyllostomus discolor, (D) Noctilio leporinus, (E) Rousettus leschenaultii, and (F) Pteropus vampyrus. Periosteal surface points up in each panel. Scale bar equals (A) 150 µm, (B) 200 µm, (C & D) 300 µm, (E) 480 µm, and (F) 800 µm. Digital slides are available at http://paleohistology.appspot.com.

Click here for additional data file.

Figure S4 Bone histology of humeri in sampled birds stained to highlight cement lines

Sections stained with toluidine blue reveal arrest lines and canals of secondary osteons, which were excluded from analysis. Representative views are from the caudal octant of (A) Phalaenoptilus nuttallii, (B) Nothura darwinii, (C) Crypturellus boucardi, (D) Crypturellus cinnamomeus, (E) Columba livia, (F) Nothoprocta cinerascens, (G) Nothocercus nigrocapillus, (H) Eudromia elegans, and (I) Tinamus major. Periosteal surface points up in each panel. Scale bar equals (A) 200 µm, (B & G) 480 µm, (C, D, F, & I) 600 µm, (E) 400 µm, and (H) 800 µm. Digital slides are available at http://paleohistology.appspot.com.

Click here for additional data file.

Supplemental Information 9 Files used in the scaling analyses

Click here for additional data file.

We thank K Ezell for assistance in molding, casting, embedding, and imaging of specimens. We also thank R Ourfalian for sharing some data from his ongoing project involving bone laminarity in the pigeon wing. J Kamilar provided consultation on the implementation of phylogenetically informed statistics. The following colleagues facilitated loans of specimens under their care: M Bucci, G Bradley, M Hall, D Steadman, and T Weber. Constructive reviews by two anonymous reviewers, J Botha-Brink, and S Swartz improved the narrative of the manuscript. Finally, we would like to thank the late D Enlow, whose seminal works in comparative hard tissue biology continue to inspire us and future generations.

Additional Information and Declarations

Competing Interests

Author Contributions

Data Deposition

The authors declare there are no competing interests.

Andrew H. Lee conceived and designed the experiments, performed the experiments, analyzed the data, contributed reagents/materials/analysis tools, wrote the paper, prepared figures and/or tables, reviewed drafts of the paper.

Erin L.R. Simons performed the experiments, analyzed the data, contributed reagents/materials/analysis tools, wrote the paper, prepared figures and/or tables, reviewed drafts of the paper.

The following information was supplied regarding the deposition of related data:

Paleohistology Repository (http://paleohistology.appspot.com)

Rhinolophus lepidus—Fig. 2A (http://paleohistology.appspot.com/Flash/Rhinolophus_H1-2_94um.html)

Macrotus californicus—Fig. 2B (http://paleohistology.appspot.com/Flash/UA3767_H1-2_99um.html)

Phyllostomus discolor—Fig. 2C (http://paleohistology.appspot.com/Flash/UA16197_H1-1_96um.html)

Noctilio leporinus—Fig. 2D (http://paleohistology.appspot.com/Flash/UA15743_H1-2_100um.html)

Rousettus leschenaultii—Fig. 2E (http://paleohistology.appspot.com/Flash/Rousettus_H1-2_98um.html)

Pteropus vampyrus—Fig. 2F (http://paleohistology.appspot.com/Flash/Pteropus_H1-2_100um.html)

Phalaenoptilus nuttallii—Fig. 3A (http://paleohistology.appspot.com/Flash/MWU264_H1-1_99um.html)

Nothura darwinii—Fig. 3B (http://paleohistology.appspot.com/Flash/UF22260_H1-2_100um.html)

Crypturellus boucardi—Fig. 3C (http://paleohistology.appspot.com/Flash/UF44840_H1-1_100um.html)

Crypturellus cinnamomeus—Fig. 3D (http://paleohistology.appspot.com/Flash/UA8699_H2-1_95um.html)

Columba livia—Fig. 3E (http://paleohistology.appspot.com/Flash/MWU256_H1-2_103um.html)

Nothoprocta cinerascens—Fig. 3F (http://paleohistology.appspot.com/Flash/UF38951_H1-2_98um.html)

Nothocercus nigrocapillus—Fig. 3G (http://paleohistology.appspot.com/Flash/UF43432_H1-2_97um.html)

Eudromia elegans—Fig. 3H (http://paleohistology.appspot.com/Flash/UF22257_H1-1_100um.html)

Tinamus major—Fig. 3I (http://paleohistology.appspot.com/Flash/UF44828_H1-2_100um.html)

Rhinolophus lepidus—Fig. S1A (http://paleohistology.appspot.com/Flash/Rhinolophus_H1-2_94um_CPL.html)

Macrotus californicus—Fig. S1B (http://paleohistology.appspot.com/Flash/UA3767_H1-2_99um_CPL.html)

Phyllostomus discolor—Fig. S1C (http://paleohistology.appspot.com/Flash/UA16197_H1-1_96um_CPL.html)

Noctilio leporinus—Fig. S1D (http://paleohistology.appspot.com/Flash/UA15743_H1-2_100um_CPL.html)

Rousettus leschenaultii—Fig. S1E (http://paleohistology.appspot.com/Flash/Rousettus_H1-2_98um_CPL.html)

Pteropus vampyrus—Fig. S1F (http://paleohistology.appspot.com/Flash/Pteropus_H1-2_100um_CPL.html)

Phalaenoptilus nuttallii—Fig. S2A (http://paleohistology.appspot.com/Flash/MWU264_H1-1_99um_CPL.html)

Nothura darwinii—Fig. S2B (http://paleohistology.appspot.com/Flash/UF22260_H1-2_100um_CPL.html)

Columba livia—Fig. S2E (http://paleohistology.appspot.com/Flash/MWU256_H1-2_103um_CPL.html)

Tinamus major—Fig. S2I (http://paleohistology.appspot.com/Flash/UF44828_H1-2_100um_CPL.html)

Rhinolophus lepidus—Fig. S3A (http://paleohistology.appspot.com/Flash/Rhinolophus_H1-2_94um_tol.html)

Macrotus californicus—Fig. S3B (http://paleohistology.appspot.com/Flash/UA3767_H1-2_99um_tol.html)

Phyllostomus discolor—Fig. S3C (http://paleohistology.appspot.com/Flash/UA16197_H1-1_96um_tol.html)

Noctilio leporinus—Fig. S3D (http://paleohistology.appspot.com/Flash/UA15743_H1-2_100um_tol.html)

Rousettus leschenaultii—Fig. S3E (http://paleohistology.appspot.com/Flash/Rousettus_H1-2_98um_tol.html)

Pteropus vampyrus—Fig. S3F (http://paleohistology.appspot.com/Flash/Pteropus_H1-2_100um_tol.html)

Phalaenoptilus nuttallii—Fig. S4A (http://paleohistology.appspot.com/Flash/MWU264_H1-1_99um_tol.html)

Nothura darwinii—Fig. S4B (http://paleohistology.appspot.com/Flash/UF22260_H1-2_100um_tol.html)

Columba livia—Fig. S4E (http://paleohistology.appspot.com/Flash/MWU256_H1-2_103um_tol.html)

Tinamus major—Fig. S4I (http://paleohistology.appspot.com/Flash/UF44828_H1-2_100um_tol.html).

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
