# Peer review of "Wing bone laminarity is not an adaptation for torsional resistance in bats"

_PeerJ, doi:10.7717/peerj.823_

## Round 0.1 · original submission · Major Revisions

· Academic Editor

Major Revisions

Reviewer 1 sent their comments by email after getting access to the supplementary information. Here they are:

--Start of Review--
The manuscript entitled "Wing bone laminarity is not an adaptation for torsional resistance in bats" uses several metrics to compare the humeral bone histology of bats and birds. This study was initially sparked by a publication noting differences in bone histology of mallards , specifically comparing their wing and leg bones. That critical study suggested a specific pattern of bone, as seen in histological section, was correlated with torsional loading of the bone. The manuscript under review tests that functional hypothesis by examining the humerus of bats, which is known to be loaded in torsion during flapping flight (based on in vivo tests by the Swartz lab), but display a different pattern of bone histology.

The authors have compiled a reasonable sample of bats that vary in size and locomotor capabilities, and used multiple methods to quantify the architecture of bones. I only have minor comments to improve the quality of this manuscript:

1) The authors should cite the work of Rensberger and Matabe 2000 in Nature. This paper includes bat bone histology.

2) Figure 1: Include taxon names and sampled elements.

3) Consider inclusion of S2 in the main document

4) The Materials and Methods section is divided by headers and it would help readability if similar headers were used in the results section.

5) It would be useful if the authors could include histological images of the humerus of pterosaurs, if available.

6) Table 2 could be moved to the Supplementary Information

7) The authors state that flight evolved separately in birds and pterosaurs, but limit their discussion of histology to pterosaurs. The authors should especially note the unique histology of Archaeopteryx (Erickson et al. 2009 PLoS One) and its histological similarity to those taxa described here. Furthermore, Erickson's work addresses growth rate of this basal taxon.

--End of Review--

The reviewers discuss a number of issues, some of them more critical than others. On some subjects, they give contrasting opinions, for instance on the importance of pterosaurs in this paper, I tend to agree with the third reviewer that they can be deleted.
The minor changes suggested by Reviewer 1, above, should be considered and, if rejected for the manuscript, explained. I am not worried about the comments by Reviewer 2.
Comments of Reviewer 3 are most serious. They fall in a number of categories, my assesment, below, follows the order that the reviewer used.
Articulation of objectives, goals. The text should be amended, but these are mostly clarification, I do not believe that major changes are needed here, but clarification would help.
'Form-function relation' comments (e.g., line 32-33, 35-36), in the manuscript these are mostly derived from the literature, and their phrasing is maybe somewhat sloppy. Please fix these, these are not major changes.
Line 73. Yan-Yin bats, very embarrassing, please fix.
Line 123 'Straighten,' please explain this. Also explain why you 'sharpen' images in photoshop, as stated in the method section. Is this really necessary?
Line 245. The issues with the use of the laminarity index are critical. Line 252 Same for the regression type.
The issue with Fig 4A is critical and needs to be addressed, I worry less about the logarithmic presentations.
Line 264 The signal to noise ratio issue needs to be clarified.
Line 271. The BMR issue also requires serious revision.
Line 156. The Zp issue requires rewriting and rethinking (line 162).
All the discussions under the validity section need to be seriously considered and modified, with the exception of the phylogenetic comment.

I hope that you will consider adjusting your manuscript and resubmitting. Upon receipt of a second version, I will send it out for review again, but I will not use Reviewer 3 for this purpose.

Reviewer 1 ·

Basic reporting

Although the manuscript seems worthy of publication, I unfortunately lack access to the Supplementary Information. As such, I cannot complete the review. Please contact me if and when the additional files are available.

Experimental design

unable to comment

Validity of the findings

unable to comment

·

Basic reporting

The article adheres to PeerJ's policies. The paper is well-written, concise and clear. The introduction, background and references are adequate for this study. All the figures are relevant and necessary. I have a few comments on the figures under general comments. This study includes all results and discussion regarding the hypothesis of this project.

Experimental design

The experimental design is sound. A larger sample size would have been preferred, but I understand the limits of obtaining large sample sizes for destructive analyses. The tests used to take small sample sizes into account appear to be good and as the authors note, more work in this field (i.e. obtaining larger sample sizes) is necessary.

Validity of the findings

The findings of this study are robust and appear to be statistically sound.

Additional comments

This article is of interest to the general reader as well as specialists in the field of bone histology. It investigates the adaptational versus phylogenetic signal in the bone microstructure of flying vertebrates. Various analyses are carried out on living birds and bats to test the hypothesis that a laminar bone tissue pattern is necessary for flapping flight. The authors ask a very interesting question, provide detailed methods and results and provide sound conclusions. Their figures are beautiful, however I would like to recommend the following:

The figures could do with some labeling. Anyone already working in the field of bone histology will find the descriptions and figure captions obvious, but those still learning would benefit from labels such as primary osteon, simple canal, EFS for external fundamental system, what looks like parallel-fibered or even lamellar bone in some subperiosteal regions etc. Secondly, as this paper focuses on the presence of laminar bone tissue at various body sizes, it is easier to read the bone histology figures in increasing body mass, rather than ordering the images phylogenetically or some other manner. For example, Figure 2's order would change to D, A, B, C, E, F and Figure 3 would be A, C, F, G, B, D, H, E, I - in this way you can follow the changes with increasing body mass far more easily and for example, in Figure 3 you can see the general increase in slower forming bone tissue at the periosteal surface much more clearly. This recommendation follows on to Table 1 - again because body mass plays such a focal role in this study - one can read the changes in the respective properties with an increase in body mass.

Otherwise, this is a good paper and the result is important and worth publishing.

·

Basic reporting

In this study, Lee and Simons ask whether there are characteristic features of the humeral midshaft that distinguish flying from non-volant vertebrates. They carry out analyses of histology and cross-sectional geometry in several bats and birds, looking to test the idea that the mechanical demands imposed by torsional loading of wing bones during flight imposes design constraints on the skeleton. They suggest that the possession of proximal wing bones of relatively large diameter, circular cross-section, and thin walls reflects one such constraint, and that increased presence of laminar bone tissue might be another, based on some previously published work.
Overall, this manuscript makes an interesting contribution to understanding of bone structural diversity, and the histological work is of exceptional quality. I particularly appreciated being able to access specimen montages at the online Paleohistology Repository. I believe, however, that there are some issues of analysis and presentation that require resolution and refinement before the text will be acceptable for publication. I’d like to see the authors do whatever is possible to tackle these in revision.
My first concern is that the authors don’t clearly articulate the questions and goals of the study in the Introduction. They note that there is little information concerning bat bone histology in general and vascularization in particular, and that this contribution provides new data from bats that vary in feeding behavior and body size. I think the Introduction will be strengthened if the authors can be clearer concerning how information about vascularization of the midshaft of the humerus can “relate form with function (both mechanical and physiological)”, as they assert (line 67). What specific function can be brought into the discussion here? How do will the measurements made by the authors reflect function in relation to form? What aspects of mechanics and physiology can be explored and tested? If bats are to be compared to non-volant mammals, what specific question(s) will be addressed? If bats are to be compared to birds, what question(s) will be addressed? The authors argue that somatic growth rate and basal metabolic rate could be critical in determining observed patterns of bone histology. The Introduction needs to lay out clearly why these three factors – mechanics, somatic growth rate, and BMR alone, are considered to be the three possible determinants of the observed traits, in a clear, succinct form. Somatic growth rate and BMR are brought into the Discussion rather late, and the paper will be stronger if the authors’ hypotheses concerning their potential relationship to bone histology is made clear up front.
Perhaps because study’s overall goals are not articulated in a clear and specific manner at the outset, the Conclusions section of the paper is rather weak. There is material here on the topic of pterosaur bones that I found irrelevant, and the overall findings of the text weren’t succinctly brought to an effective conclusion. Indeed, little or nothing was said about bats.
In several places in the text, I found that the authors’ language was imprecise or confusing to me as a reader. Given that I am probably relatively more familiar with the subject area than many readers might be, it is probably worthwhile to better clarify these points, at the least (see below).

Lines 32 – 33, “This laminar bone occurs preferentially in wing elements known to experience substantial torsional loading (de Margerie, 2002).”; This statement seems rather strong to me relative to available evidence. Our knowledge of the magnitude of torsional loading in the different bones of the wing from multiple species, particularly in flapping vs. gliding or soaring, is minimal, and mostly theoretical. I’d certainly be more comfortable with saying something about the presence of laminar bone correlating with bones hypothesized to experience elevated torsional loading, or some other somewhat more conservative claim.

Lines 35-36, “static soaring flight mode (presumably dominated by torsion), than in those with long narrow wings that use predominantly dynamic soaring (presumably dominated by bending)”; I admit to finding this confusing. Why should static soaring and dynamic soaring impose quite different patterns of loading on bone of the wing? What is it specifically about static soaring that leads one to expect high torsional loading in contrast to expectations of low torsional loads during dynamic soaring? I’d expect the mechanics of these forms of locomotion to be rather similar. Additionally, a priori, one would certainly expect that gliding or soaring of any kind would tend to induce lower skeletal stresses than flapping flight, so since these animals use flapping flight in addition to gliding/soaring, one would think that skeletal structure would need to accommodate the higher stress flapping behaviors. For one view of the relative magnitude of skeletal stresses in flapping vs. gliding, see Kirkpatrick, S. J. (1994). Scale effects on the stresses and safety factors in the wing bones of birds and bats. Journal of Experimental Biology, 190, 195–215.

Line 59, “bone histology of bat humeri”, ‘histology of bat humeri’ would be more concise, ‘bone’ is redundant with humeri

Line 73, “Yangochiroptera (microbats) and Yinpterochiroptera Yinpterochiroptera”; It is not appropriate to characterize Yinpterochiroptera as megabats. This clade includes several families that have traditionally been included in the microbats (Hipposideridae, Rhinolophidae, Megadermatidae, Craseonycteridae, Rhinopomatidae). Because these 'microbat' families are considered yinpterochiropterans, to designate the Yangochroperatans as the 'microbats' is also misleading.

Line 117-119, “Although this method was modified to offer greater descriptive precision (de Boef & Larsson, 2007; Lee, 2007), the added precision is excessive for the bat dataset because canal orientation is unambiguous.”; I think this could be deleted with no loss to the manuscript because it is not relevant here.

Line 123 - 130, I’m not sure I completely understand what “Straighten” does to the images or why it is necessary. This could be further clarified.
Line 144, “To supplement our avian sample,” – Is this meant to say “To supplement our chiropteran sample”, with the idea that the basic sample of the study comprises bats, and the supplementary materials are the avian species? Please review.
Line 160 – 161, “As estimates of torsional bone strength (Zp) are more robust when a section is circular,”; what does this mean? That polar sectional modulus is a better estimate of torsional rigidity when the beam cross-section is circular?

Lines 229-230, “Matrix organization varies with size from predominately lamellar in smaller taxa to predominately parallel-fibered in larger taxa.”; ‘smaller-bodied’ and ‘larger-bodied’ would be better than ‘smaller’ and ‘larger’.

Lines 245 – 247, “Although laminarity index of the humerus in the sampled birds varies somewhat, it is generally consistent with values reported for other birds that use flapping or static soaring modes of flight (de Margerie, 2002; Simons & O’Connor, 2012).”; It would be good to be more specific here. What are the relevant laminarity values for comparison? De Margerie's 2002 paper reports values between 0.5 and 0.8 for the humerus of mallards, and Simons and O'Connor appear to have observed laminarity values between about 0.1 and 0.8 in humeri birds that exhibit flapping and static soaring. This makes the range of results that would be generally consistent with those previously observed rather broad, and potentially consistent with patterns reflecting multiple forms of locomotion.

Lines 252-253, “Irrespective of the type of regression (OLS or PGLS) (Table 3); This appears to be the only explicit reference to the outcome of analysis of potential effect of phylogeny on the results. I would like to see this better developed, even if only in a few more sentences.

Line 253 - 254, “bats and birds of comparable size have humeri with similar torsional strength”; As noted above, I’d prefer to see this stated in the more conservative fashion – bats and birds of comparable size have humeri with similar cross-sectional geometry. If both cross-sectional geometry and bone material are similar, strength would be similar, but the material question remains open.

In addition, I find Fig 4A somewhat confusing at present. First, as also noted above, the appropriate moment arm for torsion is not be humerus length but, assuming the axis of torsion is close to the center of the bone, the radius. Second, the moment arm multiplied by the applied force produces a moment, which has units of force x distance, and it looks like that is what the authors intend to have on their Y-axis. The Y-axis is presently labeled as moment arm but has units of force x distance, so I assume it is meant to be a moment, not a moment arm. This then requires correction on two counts.

I also encourage the authors to consider a further point of presentation. When data are presented only as their log transforms, it is typically quite difficult for readers to interpret data values. It is kinder to readers to present data in their raw, untransformed format but using log scales if that is possible; then the reader can look to the axis to gain sense of the absolute magnitude of, in this case, the moment or polar moment of inertia, without need to turn to a calculator to compute an antilog. I was encouraged to adopt this practice several years into my publication career, and wish I’d been made aware of the rather earlier.

The bottom line, however, is that until the torsional moment is corrected to employ radius instead of length, it is impossible to assess whether the relationship between moment and section modulus are similar in these groups.

Line 257, “Zp is derived from the polar moment of area”; ‘calculated’ would be a better word than ‘derived’ here.

Lines 264-267, “A power analysis reveals that signal-to-noise ratio in the slopes (0.26) is small to medium, requiring nearly 400 bats and birds (a total of 800 specimens) to show significance. The large signal-to-noise ratio in the intercepts (2.28) is a concern, but we already exceeded the required sample size (6) needed to detect significance with a pooled error of 0.071.” I will preface my comments here with the caveat that I am not familiar with the particular power analysis employed here. Nonetheless, I note that this is a confusing and somewhat disturbing bit of text. First, is something backwards? The text says, I think, that the relatively small signal-to-noise ratio for the slopes requires an enormous sample for significance but the much larger signal-to-noise ratio for intercepts requires a very small sample indeed. As I said, the method isn’t familiar to me, but poor S/N requiring a very large sample and better S/N requiring only a small sample is certainly non-intuitive. Second, if the S/N for the intercepts is large and the sample size needed to detect significance is exceeded here, why is that a source of concern?

Line 268-269, “a tentative but reasonable conclusion is that torsional strength of the humerus is
comparable in bats and volant birds.” Unfortunately, for the several reasons given above, I cannot agree that this is a reasonable conclusion at this point. It may be that further analysis will yet lead to this conclusion.

Lines 271- 274, “Although basal metabolic rates (BMR) scale differently in bats and birds, the 95% confidence bands of the trendlines overlap for body sizes greater than 200 g (Fig. 4B). Therefore, low BMR only explains the paucity of primary vascular canals in small bats.”; Could this idea be fleshed out with somewhat more depth? Fig 4B shows that BMR is highly correlated with body mass in both birds and bats with somewhat different slopes, as has been previously shown (e.g. Bonaccorso, F. J., & McNab, B. K. (2003). Standard energetics of leaf-nosed bats (Hipposideridae): its relationship to intermittent-and protracted-foraging tactics in bats and birds. Journal of Comparative Physiology B 173(1), 43–53.) It’s not at clear what we are to make of this. To what extent are the authors proposing that Fig. 4B is evidence that relatively lower BMR is the cause of reduced vascularity in smaller bats? The data provided here do not exclude the possibility, but these are simply data concerning metabolic rate scaling, and there is no necessary or direct tie to bone histology. What would the authors consider better evidence? As developed presently, this line of thinking is lacking in specificity.

Experimental design

Lines 156 – 157, “polar section modulus (Zp), which is an estimate of torsional strength and average bone bending strength; First, I think it is worthwhile to give the formula for polar section modulus, because not all readers are so familiar with the topic that it’s not worth giving the formula. Second, this is only an estimate of torsional and average bending strength (more or less bone size-normalized) that can be compared among specimens if one assumes that mechanical properties of the material are identical for all specimens under comparisons. This may or may not be a good assumption. It almost certainly is not a good assumption when comparing bats and birds; Kirkpatrick’s (somewhat limited) mechanical testing of mechanical properties of compact cortical bone from bat and bird humeri showed rather substantial differences in ultimate stress, although the tests were not done at quite the same strain rate, and no other material properties (Young’s modulus, failure strain, etc.) are reported in the study. Section modulus, then, relates to strength, but isn’t strength, although there may be many papers that incorrectly make this claim. It is a measure of the resistance of a beam or beam-like structure to torsion and/or bending, but says nothing about the load at which the structure will fail, which requires information about material as well as geometry. One could say torsional or bending rigidity instead of strength and not venture and not violate the assumptions about materials or venture into the realm of strength and failure. This would be a wise and more circumspect approach.

Line 218, “yangochiropteran Pteropus vampyrus,” ; this is incorrect. Pteropus vampyrus is a yinpterochiropteran, not a yangochiropteran. The mistake carries on in line 220, where the text identifies yinpterochiropteran Rousettus leschenaultii as a yangochiropteran also.

Line 258, “which is most accurate when cross-sectional profiles are nearly circular”; This is rather vague. Exactly what is most accurate when cross-sectional shape is circular? Estimate of shear stress due to torsion from section modulus? That is true, but it's also true that estimating stress at the midshaft using the beam theory approach as a whole assumes the bone is perfectly cylindrical and loaded in pure torsion, so the assumptions go far beyond circular cross-section at midshaft. Within the broad context of assumptions made in making these estimates, the circularity of the cross-section seems, quite frankly, to be rather small potatoes. Perhaps it is worth reviewing the other assumptions for the reader as well – that the bone is a perfect cylinder of constant cross-section shape, uniform and isotropic material properties, etc. This might not be a concern me if the authors didn't want to link hypothesized bone stress to vascularity, but I think some greater caution in interpretation may be warranted here.

Validity of the findings

Line 162 – line 164, “To compare the torsional strength of bat and bird humeri, we regressed Zp against the product of body mass and total length of the humerus (moment arm)”; I find this point quite problematic. I have not examined the references cited by the authors here, and understand that these sources may be the ultimate origin of this issue. However, humerus radius, not length is the appropriate moment arm for humeral torsion; see Kirkpatrick’s paper, any engineering text on torsion in beam theory or, for example,
http://www.ecourses.ou.edu/cgi-bin/eBook.cgi?topic=me&chap_sec=02.1&page=theory.

This becomes quite an issue in the analysis, because humerus length essentially plays no role in the magnitude of torsion applied to the humerus during flight. Chord length could potentially influence torsion, magnitude of aerodynamic forces is certainly critically important.

Line 224-226, “At least for bats, phylogeny does not appear to be a major factor for the bone vascularity of humeri. Apparently, for bats as for varanid lizards (deBuffrénil, Houssaye & Böhme, 2008) and metatherian mammals (Werning, 2013), size matters.”; Perhaps. But these two species are also more closely related to each other than they are to any of the other species in the sample, are non-echolocating, frugivorous, and may share other traits. Is their similar bone histology body size-related or is it related to their shared evolutionary history, ecology, or some combination? With only specific taxa sampled, this study simply doesn't have the power to resolve this question. The authors’ conclusion suggests that it could be size per se that leads to increased vascularity in the pteropodids in the sample, but to test this idea, one would look for vascularization in the humerus of large-bodied bats from families outside Pteropodidae and avascular humeri in small-bodied pteropodids. Vampyrum spectrum, a phyllostomid with body mass typically well over 150 g, thus larger than Rousettus leschenaultii, would be one species to consider, or Chrotopterus auritus, another quite large phyllostomid. However, Rousettus leschenaultii (body mass about 110 g) is close in body mass to Noctilio leporinus (body mass over 60g) than it is to Pteropus vampyrus (body mass over 1000g), raises some concern about the robustness of this conclusion. Ig the ‘large-bodied bat’ pattern isn’t seen in N. leporinus, I’d certainly want to look at small pteropodids - something like Syconycteris australis, any Macroglossus, or even any Cynopterus, which are not too uncommon - more important. Unfortunately, in this study sample of seven species, body size and phylogeny are confounded, limiting the strength of the conclusions that can be drawn at this point. The authors could lay out alternative hypotheses, and indicate their preference for one or another, ideally with a strong rationale. But as it stands, one could argue just as convincingly that phylogeny does appear to be a major factor, and that pteropodids differ from all other bats.

Lines 276 – 286, somatic growth rates; This is a thorny subject indeed. As the authors note, whole body somatic growth rates can reflect rates of growth of individual bones rather poorly, and growth rates in wing bones in flying vertebrates are, as noted, typically higher than in most other parts of the skeleton. The argument here that bats appear to have, overall, relatively slow somatic growth and also relatively avascular bone, so it is likely that poor vascularization of the humerus reflects slow growth rates seems somewhat circular, in the absence of independent evidence concerning the rate of growth of the bone. The idea, however, is fundamentally interesting, and arises naturally from the data as presented, and could be well worth digging into more substantively. My intuition is that growth in bat wing bones is not slower than that in other mammalian bones that are quite well vascularized, including bones of the human skeleton. The wing skeleton of a large Pteropus reaches adult size over a matter of a 3-5 months, and I would be surprised to find Pteropus wing bones in the category of "among the slowest depositional rates of the various forms of bone tissue"; this seems hard to reconcile with the production of a functional skeleton for a animal with body mass over 1000 grams in a matter of weeks to months, but stranger things are observed in the biological world! A key missing link here is that between somatic growth rates and some something specific to bones.

Lines 287 – 288, “According to recent phylogenetic systematics, bats are phylogenetically bracketed by themore closely related carnivores and the more distantly related shrews (Teeling et al., 2005; Bininda-Emonds et al., 2007).”; What is meant here by ‘phylogenetically bracketed’? Shrews are certainly not the next closest major group to bats after carnivorans; ungulates would be the sister taxon to bats plus carnivores.

Lines 289 291; “If phylogenetic constraint is a major influence in bone vascularization across these mammalian taxa, then avascular to poorly vascularized long bone histology should be expressed by these taxa irrespective of their growth or basal metabolic rates. . . available evidence does suggest that the phylogenetic effect is minimal”; I think this frames phylogenetic influences in a rather crude fashion and creates a straw man. Phylogenetic constraint could certainly be an important influence without the appearance of poorly vascularized bone in the two genera for which data are presented.

The comparison with shrews is rather confusing as presented. As I understand the text, shrews are similar to bats in somatic growth rate and body size, but have more highly vascularized femora than bats and higher BMR. How does this lead to the conclusion that low vascularization in bat humeri reflects low growth rates? Shrews combine low growth rates ("approaches that of a comparably sized bat") and high vascularization. This would seem to be contradict the hypothesis that absence of laminar bone (and low vascularization of bone) in bat humeri is largely a reflection of low growth rates.

---

## Round 0.2 · Minor Revisions

· Academic Editor

Minor Revisions

I find this manuscript to be in good shape. The authors should add a discussion along the lines of item 1 and 7 of the reviewer.

Reviewer 1 ·

Basic reporting

The authors have revised the manuscript according to the suggestions presented in the first round of review. Only minor revisions to increase readability are required.

Experimental design

Reasonable

Validity of the findings

Reasonable

Additional comments

Introduction:

The introduction is surprisingly long, could be shortened, and could use some sub-headings to direct the reader. I suggest focusing on writing efficiently, succinctly, and hit the major concepts. Details can come in the methods/materials. Instead, it would be most beneficial if the authors were to condense the text so that it provides a roadmap for the reader and follows a logical progression that is easy to read. In addition, bolded sub-headings would allow the reader to follow what seems like a stream-of-consciousness and detailed approach in some parts (not all). Here are some specific parts that could be changed to increase the readability of the ms.
1) Consider breaking lines 32-34 into two sentences. Make them short and easily comparable.
2) Vary sentence length as it breaks up the monotony of the paragraphs.
3) Lines 54-58. Please be consistent. Use either body size + common name or just body size for all taxa.
4) Lines 62-86 are particularly rough.
a. Lines 62-67 probably belong in the methods section, and should at the very least be tightened up to offer readers what they need to know to get through the paper. Try to avoid bogging the intro down with details that should go elsewhere in the ms.
b. Start another paragraph at “We test…” Line 67
c. Replace lines 67-70 with “We test whether laminar bony tissue is also present in the chiropteran humerus.”
d. From here, please follow a more logical progression with leading phrases at the beginning of the sentences:
i. Laminar bone differs from other tissues in that it …
ii. If laminar bone is present in the humerus of bats, this would suggest that…
iii. Lines 74-75. This sentence just pops out of nowhere and lacks a citation. Are you referring to the Foote work?
iv. All of a sudden there is a massive intellectual jump to a biomechanical analysis. I would instead move the biomechanics statement out of this section and give it a separate sub-heading.
v. This is followed by another blunt transition to a metabolic study. Yowza. Add another bolded sub-heading. I’m not convinced Lines 37-43 are necessary in the introductory paragraph.

Methods and Materials
Here again, the text could be tightened.
Line 96, Consider revising these first sentences as it is kind of a dud and doesn’t move the paper along. Consider switching to an active voice.
The rest of the methods section is lovely.
Results and Discussion
Line 262, consider changing the sub-heading to Bats Lack Laminar Bone in their Poorly Vascularized Humeri, or some such statement that is more informative and relates directly back to the objective of the paper.
Line 281: consider adding a sub-heading “Only Some Birds Display High Laminarity in Bones Loaded in Torsion … or Retain Laminar Bone in their Humeri but Display Great Variability in Vasculature” or some such statement. Can you distill this paragraph into a headline for the paragraph?
Lines 292, 318: Make this headline much more informative about the results rather than just a place-holder.
Line 327: I would add a biomechanical-specific heading here that was informative. Field Metabolic Rate Does Not Explain the Vascular Dichotomy in Birds and Bats

Line 365: Headline, Differences in Somatic Growth Rate May Explain the Vascular Dichotomy in Birds and Bats
Conclusions
Efficient, concise, and a pleasure to read.

---

## Round 0.3 · accepted · Accept

· Academic Editor

Accept

I believe that bats are an understudied group with regard to bone histology, and that your paper can be one of the pioneering ones in this regard. Congratulations.